# Conjugates for use in peptide therapeutics: A systematic review and meta-analysis

**Ashan Wijesinghe**[1], **Sarika Kumari**[1], **Valerie Booth** [1,2]*

**1** Department of Biochemistry, Memorial University of Newfoundland, St. John's, Newfoundland and Labrador, Canada, **2** Department of Physics and Physical Oceanography, Memorial University of Newfoundland, St. John's, Newfoundland and Labrador, Canada

* vbooth@mun.ca

**Data Availability Statement:** All relevant data are within the paper and its Supporting Information files.

**Funding:** This work was funded by a Discovery Grant from the Natural Sciences and Engineering

## Abstract

While peptides can be excellent therapeutics for several conditions, their limited *in vivo* half-lives have been a major bottleneck in the development of therapeutic peptides. Conjugating the peptide to an inert chemical moiety is a strategy that has repeatedly proven to be successful in extending the half-life of some therapeutics. This systematic review and meta-analysis was conducted to examine the available literature and assess it in an unbiased manner to determine which conjugates, both biological and synthetic, provide the greatest increase in therapeutic peptide half-life. Systematic searches run on PubMed, Scopus and SciFinder databases resulted in 845 studies pertaining to the topic, 16 of these were included in this review after assessment against pre-specified inclusion criteria registered on PROSPERO (#CRD42020222579). The most common reasons for exclusion were non-IV administration and large peptide size. Of the 16 studies that were included, a diverse suite of conjugates that increased half-life from 0.1 h to 33.57 h was identified. Amongst these peptides, the largest increase in half-life was seen when conjugated with glycosaminoglycans. A meta-analysis of studies that contained fatty acid conjugates indicated that acylation contributed to a statistically significant extension of half-life. Additionally, another meta-analysis followed by a sensitivity analysis suggested that conjugation with specifically engineered recombinant peptides might contribute to a more efficient extension of peptide half-life as compared to PEGylation. Moreover, we confirmed that while polyethylene glycol is a good synthetic conjugate, its chain length likely has an impact on its effectiveness in extending half-life. Furthermore, we found that most animal studies do not include as much detail when reporting findings as compared to human studies. Inclusion of additional experimental detail on aspects such as independent assessment and randomization may be an easily accomplished strategy to drive more conjugated peptides towards clinical studies.

## Introduction

### Peptides as therapeutics

Peptides are broadly defined as molecules made up of two or more amino acids connected by a peptide bond while those with remedial properties against diseases are identified as

Research Council of Canada to V.B. (RGPIN 05154).

**Competing interests:** The authors have declared that no competing interests exist.

therapeutic peptides. Although there is no endorsed convention that differentiates between peptides and proteins, it is commonly accepted that amino acid chains exceeding 50 residues are classified as proteins [1]. The use of peptides as pharmaceuticals began in the 1920s following the massive success of insulin in treating diabetes [2]. Moreover, in just the last decade, 18 peptide drugs were approved by the U.S. Food and Drug Administration (FDA) bringing the total number of peptide drugs licensed for use in the USA to 60, while ~600 others have been studied in clinical and preclinical trials [3–5]. Interestingly, while many of these trials investigated the efficacy of peptide therapeutics for use in the fields of oncology, cardiovascular disease, and metabolic disease, a large fraction of these peptides did not go into development and subsequent clinical use [2].

Therapeutic peptides have been categorized as native, analogue, and heterologous, according to their relationship to endogenous peptides [2]. Native peptide drugs have the same sequence as naturally occurring peptides and can be extracted from natural sources or produced synthetically. Analogues are versions of native peptides modified to exhibit enhanced drug properties like increased half-life and improved target specificity. Conversely, heterologous peptides are not designed using natural peptide templates, rather, they are discovered through approaches such as phage display, a high throughput screening technique developed to screen large libraries of peptides [3, 6]. Peptides of all these categories are, however, designed to amplify the qualities that make them good therapeutics.

A few innate qualities of peptides that make them appealing as potential therapeutics are their small size, predictable metabolism, and target specificity. As compared to antibody-based drugs, the largest of polypeptide therapeutics currently in use, peptides are up to one order of magnitude smaller [7]. This smaller size makes the production of peptide therapeutics easier and less expensive. Peptides are also advantageous over small molecule drugs in that they possess natural target specificities where tight binding is often limited to a single binding partner. In therapeutics, this quality greatly reduces the incidence of unfavourable side-effects due to off-target binding [8]. Moreover, some peptides bind to G-protein coupled receptors where a potent, amplified cascade of reactions is initiated upon binding. This quality allows for the administration of low doses to induce substantial effects [9]. Additionally, since peptides are biological molecules, they are readily metabolised into compounds that are rarely toxic to the body. These metabolites follow the same elimination routes as do endogenous metabolites, and therefore, their accumulation is not common [3]. Since metabolite accumulation and toxicity is often a reason for drugs to fail clinical trials, this gives peptide therapeutics an implicit advantage over other drug candidates [3]. Overall, peptides make for well-suited therapeutic interventions in natural pathways since much of the body's physiological functions like glucose uptake, water retention, energy metabolism, growth, and regeneration, etc., are governed by peptides that are intrinsic signalling molecules. However, as with any molecule, biological moieties like peptides face several limitations as therapeutics.

## Limitations to peptides as therapeutics

Intrinsic limitations of peptides have impeded the development of long-lasting, effective peptide drugs. The two major constraints in peptide drug development are their poor oral bioavailability and short half-life in the bloodstream [10]. Peptide drugs, like dietary peptides, are susceptible to digestive enzymes along the gastrointestinal tract. Even when peptides do make it past the stomach, intestinal impermeability to molecules of their size limits entry to the systemic circulation [3, 11]. Peptide therapeutics are consequently often limited to delivery via injection, an administration route that is still well suited to the management of acute diseases

[3]. Much research is, therefore, being carried out to optimize the residence time of such therapeutics in the bloodstream [11].

Once in the bloodstream, studies have shown that peptides without special modifications often only last minutes to a couple of hours before they are cleared by proteolysis or renal filtration [12]. Plasma clearance is generally dependent on two main peptide characteristics: size and surface charge. Strategies that are employed to increase plasma half-life typically manipulate one or both of these properties [13]. Therapeutic molecules that are smaller than the threshold for renal filtration, which is thought to be ~70 kDa, are more likely to undergo faster renal clearance as compared to those that are larger [14]. Similarly, positively charged molecules are readily cleared by the kidneys due to their attraction to the negatively charged basement membrane of renal tubules [14]. Proteolysis, on the other hand, can occur both within organs and in the bloodstream [13]. A recurrent mechanism used to avoid extracellular proteases is to identify and alter their target site in therapeutic peptides [3]. This is commonly done by incorporating D-amino acids, which are not susceptible to degradation by endogenous proteases, into these target sites [7]. Manipulating peptide size and charge can, therefore, aid in evading renal clearance while stabilizing the peptide structure can help evade extracellular proteolysis.

Bypassing proteolysis in the liver depends on the peptide's fate following receptor-mediated uptake. Peptides that enter hepatocytes in this manner are contained within endosomes where they undergo lysosomal degradation [15]. Certain proteins and peptides are, however, rescued by receptor-mediated recycling that sorts these moieties away from the endosome and returns them to the cell surface to continue in circulation. Studies have shown that attaching peptides to moieties that are capable of evading lysosomal degradation in this manner results in lengthened plasma half-life [12]. There are, therefore, a number of variables to consider when exploring how to extend peptide half-life in the body.

## Conjugation and its benefits

Peptide conjugation is a broad approach that encompasses the attachment of chemical moieties to peptides for several reasons including to improve their drug and diagnostic properties. Conjugates used in peptide therapeutics can either be nonbiological molecules like polyethylene glycol (PEG) or biological molecules such as lipids, sugars, and proteins [1, 3, 7, 9, 10, 13]. PEG is one of the most prominent nonbiological conjugates used and has been studied extensively. There are currently about 15 FDA approved PEGylated peptide drugs in the market [3]. That said, PEG is not metabolised nor excreted as efficiently as biological molecules are. Therefore, immune reactions and renal effects have been observed due to the toxic accumulation of PEG metabolites [9, 16]. Consequently, recent research has been exploring alternatives to PEG as a conjugate. With the advancement of recombinant protein production technologies, recombinant peptides too have been incorporated into the range of moieties conjugated to peptide drugs. Unlike with PEG, these conjugates follow the same metabolism and elimination routes as do their therapeutic and endogenous counterparts making them much suited for peptide therapeutics. Another significant area of research includes the study of peptides attached to natural protein domains with inherent bioactivity such as human serum albumin (HSA) and transferrin proteins. In general, studying the pharmacokinetics, or the movement and eventual fate of the conjugates within the body, is of primary interest as it reflects how well the peptides will fare in terms of plasma clearance [14].

In the search for conjugates with favourable characteristics, researchers have observed that attaching moieties capable of increasing the peptide's size and/or altering its charge has potential to successfully extend plasma half-life. Attaching a conjugate that is relatively large like

PEG can increase the MW of the peptide by 2–40 kDa allowing it to evade kidney filtration [7]. Similarly, attaching a conjugate that is negatively charged helps avoids renal clearance as detailed in the previous section [13]. This does, however, raise concerns about how the peptide-conjugate complex's interactions with the targeted substrate might change if the overall charge is altered. Conjugation can also prevent hepatic metabolism of peptides when the conjugate can return the peptide to the hepatic cell surface due to intrinsic bioactivity as demonstrated by HSA and certain antibodies [15]. Moreover, many natural protein domains used for this purpose are large enough in size to allow conjugated peptides to evade renal clearance [15]. Research has also shown that HSA can function as a carrier for other molecules. This means that conjugating peptides to HSA-binding molecules like lipids can effectively convey the same properties as directly binding to HSA [12, 13, 15].

Additionally, it has been observed that extracellular enzymatic degradation too can be minimized by conjugation [3]. Since endogenous proteases are liable to act on unstructured regions of peptides, conjugates that stabilize the structure of the therapeutic peptide can help escape enzymatic degradation [3, 11]. Further, addition of conjugates can confer a steric shielding effect against proteases and peptidases [3]. Glycosylation and encapsulation of the therapeutic peptide are a few such forms of conjugation that are known to confer protection from extracellular proteases [13]. On the whole, it is apparent that conjugation can be successful in prolonging the half-life of therapeutic peptides in the bloodstream.

## The purpose of this systematic review and meta-analysis

Peptides have the potential to become successful therapeutics if the challenge of prolonging their half-life is met. With conjugation being identified as an effective strategy in extending plasma half-life, studies investigating potential conjugates have become more prominent in the past several years. Studying conjugated peptide behaviour also provides insight for drug producers to design and synthesize effective, long-lasting, and less-toxic therapeutic interventions. In just the past decade, 30% of the peptide drugs entering the clinical development stage were conjugated and 40% of these targeted G-protein coupled receptors as molecular targets [2]. Alongside conjugation, scientists are also looking at peptide backbone modification, and tertiary and quaternary structure modification [13]. However, for this systematic review, we were interested in the data surrounding conjugation as it is by far the least expensive and least complicated strategy employed to extend the half-life of peptides. Furthermore, since there have been large technological advances in the production of peptides by both recombinant expression and chemical synthesis recently, we chose to limit the study to just the last five years.

The main purpose of this systematic review and meta-analysis is to compare the various non-specific conjugates in terms of their contribution to the half-life of therapeutic peptides in the bloodstream as observed in animal studies. With all the above considerations in mind, we aimed to answer the question "**within the last 5 years, which non-specific conjugates for therapeutic peptides has led to the greatest peptide half-life in the bloodstream of animals?**" In attempting to answer this question in our research, we are hopeful that this systematic review and meta-analysis would contribute to expediting therapeutic design and testing processes, so that more conjugated peptides may be approved as therapeutics in the near future.

## Methods

The methodology detailed here and the eligibility criteria pertinent to this systematic review and meta-analysis were decided upon in advance and documented in the International

**Table 1. List of search terms used in developing search strategies.**

| Search Terms | | |
| --- | --- | --- |
| **Peptide-based therap**[*] | **Nanoparticle**[*] | **ELPylation** |
| Peptide therap[*] | Dextran | Elastin-like polypeptide |
| Peptide drug | Unnatural amino acid | N-glycosylation |
| Therapeutic peptide[*] | PASylation | Polysialylation |
| Anticancer peptide[*] | HESylation | Conjugate |
| Peptide conjugate[*] | HA conjugation | Half-life |
| PEG | HAylation | Half life |
| Polyethylene glycol[*] | XTEN | |
| Liposome[*] | PEGylation | |

Note

[*] = truncations.

Prospective Register of Systematic Reviews (PROSPERO) (Registration # CRD42020222579; S1 Data). Note: the term 'biologically inert' used to describe the conjugates being studied was switched out for 'non-specific' as per the reasoning outlined in the Eligibility criteria section below. The registered protocol, and this paper, follows the Preferred Reporting Items for Systematic Reviews and Meta-Analysis (PRISMA) guidelines in conducting and reporting this Systematic Review and Meta-Analysis (SRMA) [17].

## Search strategy

We identified PubMed, Scopus, and SciFinder as citation databases suitable for our literature search as they cover a range of scientific disciplines surrounding biomedical sciences and medicinal chemistry. Based on these citation databases, a custom search strategy was developed incorporating the search terms given below (Table 1). A preliminary search revealed that a large fraction of the studies extracted by these search terms concerned vaccines and antibodies, which are not relevant to the scope of this study. To exclude these studies, we included the following NOT terms: antibodies; vaccine[*]; vaccination. Where possible, the search was also limited by publication language (English) and date range (1st Sept 2015 to 1st Sept 2020) to reflect the characteristics highlighted in the eligibility criteria (Table 2). The individual search strategies used for each database can be viewed in the supplementary information (S1 Table). Systematic review specialists from the Memorial University of Newfoundland libraries were consulted to optimize the search strategy and ensure that it would capture all relevant studies. The databases were last searched on 11 Dec 2020.

## Eligibility criteria

When screening the studies captured in our search, we ensured that the studies chosen met the pre-specified eligibility criteria (Table 2). All the studies chosen presented plasma half-life data in the form of a table or figure. The peptides studied were no larger than insulin in size (5.8 kDa) and were intravenously (IV) administered. This ensured that the data would not be skewed by peptides that were inherently able to avoid plasma clearance due to larger size. Similarly, peptides that were administered via non-intravenous routes, like subcutaneous (SC) or intraperitoneal (IP) injections, were excluded as these routes involve the slow release of therapeutic into the bloodstream, thereby increasing the effective half-life of the peptide [18]. Peptides with minor modifications like N & C terminal capping, or modifications required to cyclize peptides were included. Peptides that were modified to the extent where the molar

**Table 2. Eligibility criteria upon which studies were chosen for the systematic review.**

| Inclusion Criteria | Exclusion Criteria |
|---|---|
| Must be peer-reviewed primary literature. | Must not be a case report, abstract, review, note or conference letter, book chapter or patent |
| Must be a therapeutic peptide | Must not be a non-therapeutic peptide |
| Must be intravenously administered | Must not be non-intravenously administered |
| Must study healthy animals | Must not use diseased animals |
| Must have data on half-life | Must not study a vaccine-related conjugate |
| Must consist of L-amino acids | Must not consist of D-amino acids |
| Must be smaller than insulin in size (5.8 kDa) | Must not be larger than insulin (5.8 kDa) |
| Must be a non-specific, biologically inert conjugate | Must not be a targeted, or specific conjugate |
| May contain minor modifications | Must not be overly modified |
| Must be an English language publication | Must not be a non-English language publication |
| Must have been published between 1st Sept 2015 and 1st Sept 2020 | Must not have been published prior to 1st Sept 2015 or after 1st Sept 2020 |

mass of non-peptide moieties approached or exceeded the molar mass of peptide moieties, as in the case of daptomycin, were excluded. Peptides that were composed of D-amino acids were excluded as they can avoid enzymatic degradation via endogenous proteases and peptidases, thereby inherently increasing plasma half-life.

The conjugates included were not restricted in terms of size or molecular class, but conjugates that specifically targeted a cell or tissue type in general were excluded. However, there were some conjugates that were on the border of specific vs. non-specific, namely, high-density lipoproteins (HDL) and HSA. We chose to include these moieties as they are transport molecules with targets found abundantly throughout the body, and thus quite different in character compared to truly specific conjugates like target-specific antibodies. HDL participates in reverse-cholesterol transport, a process in which excess peripheral cholesterol is picked up and delivered to the liver [19]. HDL-conjugated drugs, therefore, interact with a vast variety of tissues in the periphery before reaching the liver. As such, HDL is considered a suitable conjugate for therapeutic peptides whose target/s may be encountered prior to metabolism in the liver [20]. HSA, too, functions similarly in that it transports hormones, vitamins, enzymes, etc. throughout the body. Additionally, HSA-conjugation increases solubility in aqueous media and helps avoid renal clearance [15].

Lastly, we also ensured that only peer-reviewed primary literature was included in this review, thus excluding case reports, abstracts, reviews, conference notes, and patents. The studies included were English language publications made available online between 1st Sept 2015 and 1st Sept 2020.

## Study selection and extraction

The citations captured by the systematic searches were imported into Covidence, a systematic review management software [21]. Covidence automatically detects and removes duplicates upon importation. The duplicate citations excluded at this stage were manually reviewed to ensure they were true duplicates. The rest of the studies were screened in two stages to ensure they fit the eligibility criteria precisely.

In the first stage of screening, only the titles and abstracts were examined with respect to the pre-specified eligibility criteria (Table 2). Screening was conducted independently by two out of three reviewers, with papers randomly assigned to each reviewer. Each study required two votes

to move further: two 'yes' votes to proceed to the next stage, or two 'no' votes to be eliminated. Any conflicts were overruled by a third reviewer's vote or, alternately, consensus was reached by discussion among reviewers. If the title and abstract did not contain adequate details, or if the reviewers were unsure of its eligibility, the study was voted 'yes' and moved on to the next stage of screening. The second stage of screening involved assessing the full text of the articles against the eligibility criteria (Table 2). This process too was conducted independently by two reviewers. The third reviewer cast the deciding vote to resolve any disagreements between reviewers.

A specialized data extraction sheet based on the Cochrane Consumers and Communication Review Group's data extraction template was devised to extract information for the review [22]. This extraction form sought information like the aim of the study, study design characteristics, animal model characteristics, intervention characteristics (peptide therapeutic, size, administration route, and controls used), conjugate characteristics (conjugate and conjugate size), conjugated peptide size, outcome measure (method and half-life recorded), study funding sources, and possible conflicts of interest. The primary outcome sought was the half-life recorded for each peptide therapeutic when conjugated vs unconjugated. Only data available in the papers were extracted. In order to avoid non peer-reviewed data, unpublished details were not sought-after or included in the SRMA.

## Data analysis

**Meta-statistics.** The data extracted was qualitatively and quantitatively analysed (see S1 Data for an excel spreadsheet with the number values and formulae used). Only studies that included both experimentally determined control values and standard deviations for all measurements were meta-analysed via a random effects model. For these studies, we compared the absolute mean difference of the peptide half-life (in hours) when conjugated vs unconjugated. Since all the included studies presented outcomes on the same scale (i.e., half-life in hours/min), there was no need for more complex standardised or normalized means. However, in experiments where a single control cohort was shared amongst multiple comparisons, the true control cohort per comparison was adjusted as per equation 2 in Vesterinen *et al.* [23]. The standard error of the effect size was then calculated by combining the standard deviations of the experimental and corrected control groups as detailed in equations 5 & 6 [23].

Since this review includes many different peptides, doses, animal models, and sample sizes, the true effect size varied from study to study. The random effects model accommodated these differences when determining the combined weighted effect and total variability [24]. This model accounts for both intra-study and inter-study variance, where the intra-study variance is calculated using the Tau-squared ($\tau^2$) statistic and inter-study variance is adapted from the individual studies [23–25]. Assigning weights to studies ensures that more emphasis is placed on studies that carry more information. Therefore, studies with a sample size of 5, for instance, are given more weight than those with only 3 subjects [25]. We chose to weight the studies by the inverse of the sum of intra-and inter-study variance, a value that is derived based on corrected sample size and standard deviations [23]. This approach is similar to assigning weightings proportional to sample size but is more nuanced in that it minimizes the variance of the combined effect [24]. Additionally, we also tested for heterogeneity using the Higgins *et al.* [26] $I^2$ model. This statistic is a ratio of intra-study variance to inter-study variance that highlights any important differences (if present) between the studies that may influence the outcome [26]. Lastly, a z-test was conducted to determine the statistical significance of the overall effects [24]. The statistics involved in the random effects model including combined effect sizes, total variance, heterogeneity, and the z-test were calculated using equations 11.2, 11.3, 12.2 and 12.5–12.14 adapted from Borenstein *et al.* [24].

**Sensitivity analysis.** A sensitivity analysis was used to assess each study's contribution to the heterogeneity of the combined effect size [27]. To do so, the combined effect size was recalculated while excluding individual studies in the meta-analysis. Any deviation from the overall combined estimate indicated the extent to which the study excluded contributed to the heterogeneity observed, which was used to make inferences about the peptide and conjugate involved [27].

**Risk of bias within studies.** A risk of bias (RoB) analysis was conducted for each study in an anonymized standardized manner by two reviewers. RoB analyses attempt to identify any systematic errors in the methodology of a study with the potential to have affected the results [28]. The Systematic Review Centre for Laboratory Animal Experimentation (SYRCLE) RoB tool was used to assess the risk of bias and reporting quality of the studies in our review [29]. RoB was assessed across nine domains using detailed signalling questions that were adapted from the Hooijmans *et al.* [29] guide. For entry 2: Selection bias–Baseline characteristics, it was decided that comparable baseline characteristics include age, sex, and health of the animals.

## Results and discussion

The systematic search conducted to address the research question regarding which conjugate led to the greatest peptide half-life in the bloodstream of animals yielded a total of 845 studies: PubMed = 631, Scopus = 82 & SciFinder = 132 (Fig 1). To ensure that only data of high quality

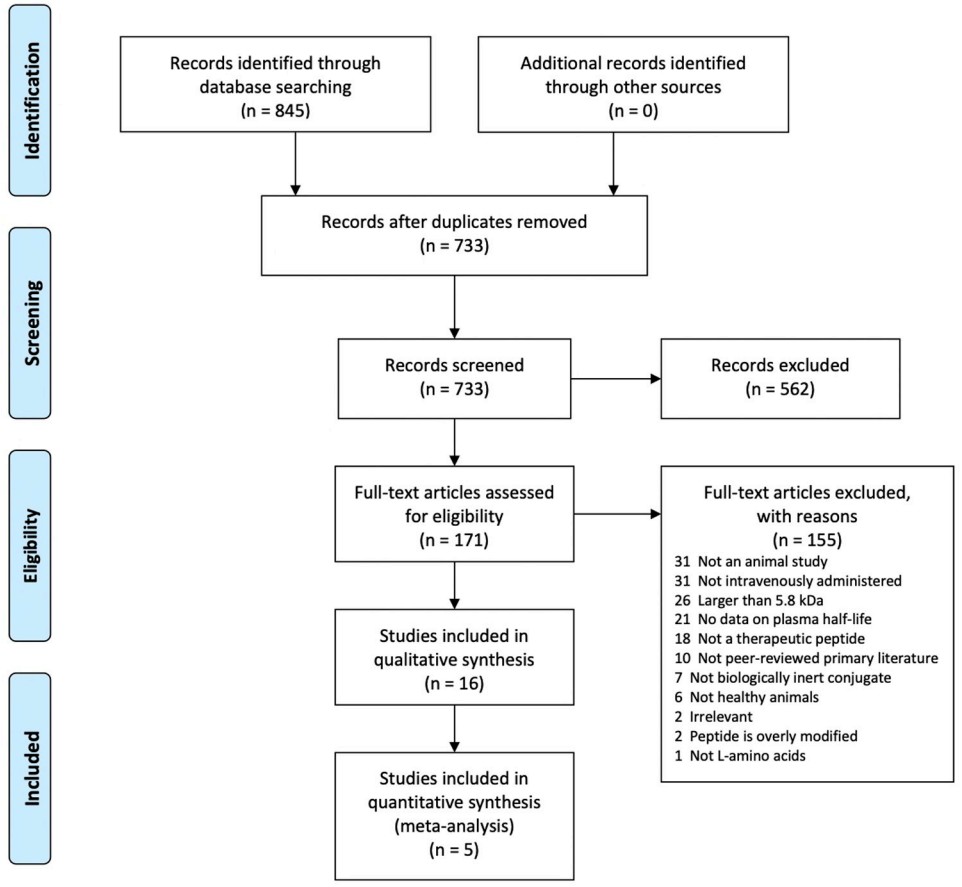

**Fig 1. Flow diagram indicating the breakdown of the literature search results.** This flow chart of the selection process was prepared following the template provided by Liberati *et al.*, 2009 [31].

were included in the SRMA, 'grey literature' or research not disseminated through peer-reviewed academic journals, were excluded [30]. The peer-reviewed, primary literature captured in the individual searches were imported to Covidence which automatically removed duplicates [21]. Of the 733 records left to screen, a further 561 records were excluded during the Title and Abstract screening stage as they did not fit the pre-specified inclusion criteria (Table 2). The full texts of the remaining 172 records were then screened to identify 16 studies which matched the selection criteria. The 156 studies excluded at the full text screening stage were sorted into exclusion reasons depicted in Fig 1. It was common for the studies to be excluded for multiple reasons; however, Covidence only keeps track of one per study. A few common reasons for exclusion were the large size of the therapeutic peptide that exceeded the cut-off (Table 2) and administration via SC or IP routes rather than IV. Of the 16 studies included in the review, only 5 studies were eligible for the application of meta-statistics. This is because performing a meta-analysis requires well-defined, experimentally determined controls as well as a measure of standard deviations, which were not available in the remainder of the studies.

## Data overview

The 16 studies included in the qualitative analysis captured a wide variety of therapeutic peptides with a range of physiological effects. Details of all 16 studies are given in the supplementary material (S2 Table), while the studies that included comparable controls were further analysed as detailed in the next section. Some of the peptides captured in the group of 16 were well-studied native peptides like insulin and glucagon-like peptide 1 (GLP-1) [32–36]. Others were novel, heterologous peptides like PN-2921, an interleukin-6 binding peptide derived from a helix-loop-helix scaffold identified through phage display [37]. The most abundant, however, were peptide analogues of native peptides such as 22A, a mimetic of apolipoprotein A optimized for conjugation, and Hirulog, a hirudin mimetic [38–40].

Along with the wide range of peptides, a diverse suite of conjugate moieties was encountered in the 16 studies included (S2 Table). Thirty five percent of these studies used PEG or a modified version of PEG as a conjugate, while another 41% of the studies used lipids including fatty acids, cholesterol, and phospholipids, either on their own or conjugated to PEG. While PEG still makes up a significant portion of the conjugates used, seeing more variety is a promising observation since PEG is known to induce immunogenic responses, among other complications [9, 16]. In addition to the lipid conjugates, there were several other biological conjugates like chondroitin (CH) and heparosan (HPN) which are glycosaminoglycans, or linear chains of repeating disaccharides. Also captured in the search were recombinant protein conjugates such as the repebody (repeated scaffolds of an antibody protein) seen in the Kim *et al.* [32] study and the acylated peptide tag used in the Zorzi *et al.* [41] study. It is important to note that the repebody used in the Kim *et al.* [32] study, while derived from an antibody, does not express the typical target specificity of true antibodies and is, therefore, considered a non-specific conjugate as per our eligibility criteria.

The animal models used in the studies comprised of varying strains of mice (BALB, ICR, and CD-1), rats (SD, Wistar), and monkeys (S2 Table). In most of the studies, the sex of the animals was specified, with many being male animals. Hence, caution must be taken when extrapolating results from a unisexual sample since the affects may not always be the same in the opposite sex [42]. The preferred methods for measuring the amount of peptide present in the bloodstream, and hence the half-lives, were enzyme-linked immunosorbent assays (ELISA) or some form of liquid chromatography-mass spectrometry (LC-MS).

## Comparison of effect sizes

In comparing the absolute mean difference (effect size) of half-lives of the conjugated peptides to those of the control groups (unconjugated), it is evident that conjugation was generally successful in increasing the plasma half-life of the therapeutics (Table 3). A range of half-lives from as short as 6 minutes to as long as nearly a day and a half were reported, with the majority of the conjugated peptides exhibiting a half-life extension of several hours. In regards to the control values needed to calculate the effect size for conjugation, four situations were encountered: 1) some studies presented experimentally determined control half-lives which were used in calculating the effect size; 2) other studies did not include experimental controls, but compared their peptide half-lives to control values available in the literature, which we emulated (indicated by a dagger (†) superscript in Table 3); 3) a third group of studies did not include any comparable half-lives, yet, when possible, we calculated their effect sizes by comparing against control values from other papers captured within our systematic review which studied the same peptide (indicated by a double dagger (‡) superscript in Table 3); 4) lastly, studies that could use neither of these options were excluded entirely from Table 3. It seems likely that some studies did not include experimentally determined controls because the short half-life of the unconjugated peptide was difficult to measure.

The mean half-lives of conjugated peptides are seen to increase overall as indicated in Table 3. There are, however, a few exceptions observed as in the cases of Fawaz *et al.* [38] and McVicar, Rayavara & Carney [47]. Note that the half-life of the control peptide for Fawaz *et al.* came from Tang *et al.* [40]. In the McVicar, Rayavara & Carney study, TP508 was conjugated with 4 different sizes of PEG, the smallest of which (PEG5k) exhibited a slight decrease in half-life (Table 3).

The smallest increase in half-life when conjugated ($t_{1/2}$ = 0.10 h) was observed in the Fu *et al.* [45] study of leuprolide (LEU) conjugated with PEG2k (Table 3; Fig 2). When comparing this study to others that involved PEGylated peptides, it was noted that the plasma half-life of the peptides was shorter when the attached PEG moieties were smaller. This effect is observed in both the Fu *et al.* [45] and McVicar, Rayavara & Carney [47] studies where the mean half-life of the conjugated peptide increased as the size of PEG increased. In contrast, the highest comparative effect was recorded in the Ichikawa *et al.* [35] work conjugating GLP-1 with 50 kDa heparosan (Fig 2). In this instance, the conjugated peptide conferred an absolute mean difference of 33.55 h (Table 3). Fig 2 depicts the range across which conjugation increased the mean plasma half-life of the peptides captured in our study. Thus, the data in Fig 2 illustrates how the choice of conjugate should be driven by the desired half-life for the desired biological effect of the peptide therapeutic. It is, however, also important to note that the peptide itself can affect the therapeutic's interactions in the body, and can, therefore, play a role in determining its half-life.

## Meta-statistics

From the studies that were captured in our search, only 5 contained both experimentally derived control values and standard deviations, which were included in our random effects meta-analysis model [23–25] (Table 4). Note that Fu *et al.* [45] recorded details for two comparison groups, therefore, a total of 6 comparisons were included (Table 4).

The weighted effect sizes (WES) of each individual study and the combined effect estimate indicates a positive effect that favours the conjugated peptide treatment (Table 4; Fig 3A). However, since the confidence intervals (CI) spread beyond the line of null effect, as seen in Fig 3A, they do not indicate statistically significant results. The same is reflected in the z-test analysing overall effect (z = 1.34) with a two-tailed p-value of 0.18. These effects may be

**Table 3. Reported experimental half-lives compared as % increase and absolute mean differences.**

| Study | Therapeutic Peptide | Conjugate | Mean Control (h) | Mean Experimental (h) | Effect Size (h)[a] |
|---|---|---|---|---|---|
| Bak 2020 [33] | GLP-1 | 16HSA | ~0.05[†] [43] | 8.4 | 8.35 |
| | | 19HSA | | 7.4 | 7.35 |
| | | 28HSA | | 8 | 7.95 |
| Chen 2016 [44] | Exendin-4 | tEB (Evan's Blue dye derivative) | 0.064 | 0.341 | 0.28 |
| Fawaz 2020 [38] | 22A (apo A1 mimetic peptide) | 1-palmitoyl-2-oleoyl-sn-glycero-3-phosphocholine (POPC) | 3.8[‡] [40] | 3.3 | -0.50 |
| | | 1,2-dimyristoyl-sn-glycero-3-phosphocholine (DMPC) | | 3 | -0.80 |
| | | 1,2-dipalmitoyl-sn-glycero-3-phosphocholine (DPPC) | | 3.3 | -0.50 |
| | | 1,2- distearoyl-sn-glycero-3-phosphocholine (DSPC) | | 3.3 | -0.50 |
| Fu 2020* [45] | LEU (leuprolide) | PEG2K | 0.43 ± 0.025 | 0.53 ± 0.032 | 0.10 |
| | | PEG5K | | 1.28 ± 0.64 | 0.85 |
| Fukushima 2019 [34] | Insulin | Chondroitin-C3-GlyA1 | 0.167 | 5.6 | 5.43 |
| | | Chondroitin-C3-LysB29 | | 3.4 | 3.23 |
| | | Chondroitin-C3-GlyA1/LysB29 | | 7.5 | 7.33 |
| | | Chondroitin-C6-GlyA1 | | 5.6 | 5.43 |
| | | Chondroitin-C6-LysB29 | | 2.2 | 2.03 |
| | | Chondroitin-C6-GlyA1/LysB29 | | 9.4 | 9.23 |
| | | Chondroitin-C11-GlyA1 | | 4.8 | 4.63 |
| | | Chondroitin-C11-LysB29 | | 4.9 | 4.73 |
| | | Chondroitin-C11-GlyA1/LysB29 | | 14 | 13.83 |
| | | Heparosan-C3-GlyA1 | | 7.3 | 7.13 |
| | | Heparosan-C3-LysB29 | | 6.1 | 5.93 |
| | | Heparosan-C3-GlyA1/LysB29 | | 16.9 | 16.73 |
| | | Heparosan-C11-GlyA1 | | 5.6 | 5.43 |
| | | Heparosan-C11-LysB29 | | 6.9 | 6.73 |
| | | Heparosan-C11-GlyA1/LysB29 | | 12.9 | 12.73 |
| | | Chondroitin-C3-GlyA1 (rats) | 0.075 | 5.7 | 5.63 |
| | | Heparosan-C3-GlyA1 (rats) | | 8 | 7.93 |
| Ichikawa 2018 [35] | GLP-1C | Chondroitin70-ethylenediamine- N-[λ-maleimidododecanoyloxy]-sulfosuccinimide | ≤0.05[†] [43] | 32.9 | 32.85 |
| | | Chondroitin90-ethylenediamine- N-[λ-maleimidododecanoyloxy]-sulfosuccinimide | | 25.3 | 25.25 |
| | | Heparosan50-ethylenediamine- N-[λ-maleimidododecanoyloxy]-sulfosuccinimide | | 33.6 | 33.55 |
| Kim 2019 [32] | GLP-1 | HSA-specific repebody | ~0.05[†] [43] | 10.7 | 10.65 |
| Lear 2020 [46] | PYY2 (peptide tyrosine tyrosine) | PEG/Fatty acids (S11) | 1.21 | 14.4 | 13.19 |
| Liu 2015* [39] | Hirulog | Stearic acid (acylated Hirulog) | 0.226 ± 0.043 | 3.54 ± 0.97 | 3.31 |
| McVicar 2017 [47] | TP508 (508–530 of human prothrombin) | PEG5k | 0.228 | 0.19 | -0.04 |
| | | PEG20k-Cys14 | | 1.17 | 0.94 |
| | | PEG20k-N terminal | | 1.55 | 1.32 |
| | | PEG30k | | 4.3 | 4.07 |
| Tan 2017* [48] | Thymopentin (TP5) | Myristic Acid (MA) | 0.022 ± 0.004 | 1.75 ± 0.72 | 1.73 |
| Tang 2017* [40] | 22A | sHDL (syntheticHDL) | 3.8 ± 9.6 | 6.27 ± 16.6 | 2.47 |
| Zorzi 2017* [41] | UK18 (an inhibitor of urokinase) | Peptide-fatty acid (FA) tag | 0.3 ± 0.02 | 7.4 ± 0.2 | 7.10 |

[a]Effect Size indicates the Absolute Mean Differences.

[†]Literature values used for half-life of control peptide.

[‡]Control values taken from a different study captured within this systematic review

*Meta-statistics applied.

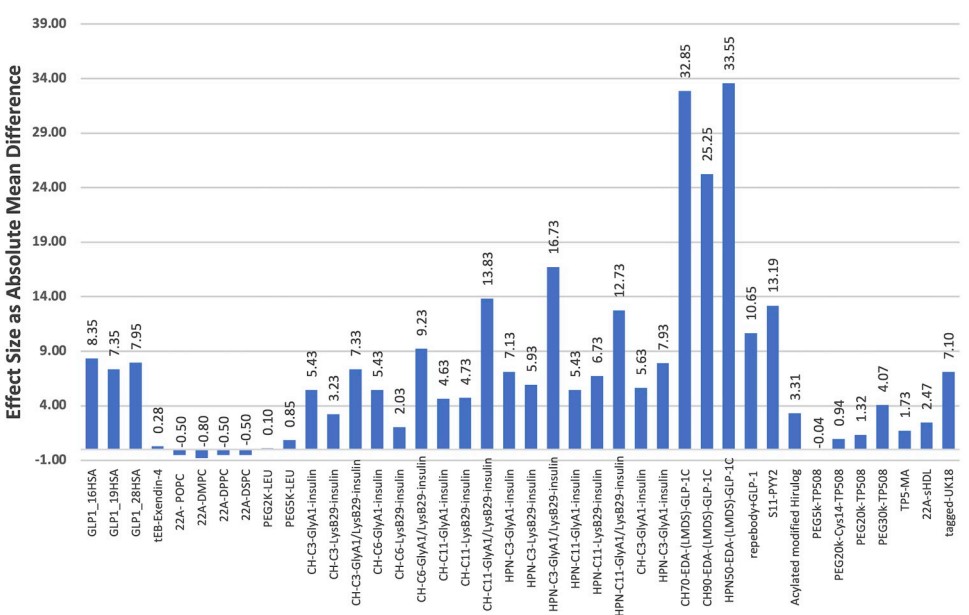

**Fig 2. The effect size of conjugation represented as absolute mean differences of plasma half-life between conjugated and control (unconjugated) peptides.**

observed due to the heterogeneity ($I^2$) between the studies which was reported to be 99.9% (indicating very high heterogeneity) [26]. High heterogeneity, however, is expected as we are comparing effect sizes of studies that work with different therapeutic peptides, conjugates, sample sizes, sexes, and drug doses. Moreover, the inter-study variance, or Tau-square ($\tau^2$), was calculated to be 19.54 for these studies. The square-root of $\tau^2$ is an estimate of the standard deviation of underlying effects across the studies analysed. Therefore, the larger $\tau^2$ is, the higher the underlying heterogeneity of the set of studies. It is, howbeit, noteworthy that if the number of studies is very small, then $\tau^2$ will have poor precision [24]. Consequently, $I^2$ and the CI which are derived using total variance will, therefore, also carry forward the poor precision of $\tau^2$.

A second Forest Plot was graphed to compare studies that used similar conjugates in an attempt to minimize the heterogeneity observed when comparing the entire collection of peptides (Fig 3B). Of the eligible studies detailed in Table 4, two used fatty acids as conjugates (Liu *et al.* [39] & Tan *et al.* [48]) which were analysed via a second random effects model. As expected, the heterogeneity score decreased to $I^2 = 88.3\%$. This, however, still falls within the classification of high heterogeneity [26]. The high $I^2$ value in this case may be due to the different peptide drugs and the different strains of rats (Sprague-Dawley rats and Wistar rats

**Table 4. Meta-statistics calculated for the studies included in the meta-analysis.**

| Study Name | Peptide | Conjugate | Effect Size | Variance (intra-study) | Weight | Weighted Effect Size | Variance (total) |
|---|---|---|---|---|---|---|---|
| Fu 2020(a) [45] | LEU | PEG2K | 0.10 | 0.02 | 0.086 | 0.01 | 11.56 |
| Fu 2020(b) [45] | LEU | PEG5K | 0.85 | 0.42 | 0.084 | 0.07 | 11.96 |
| Liu 2015 [39] | Hirulog | Stearic acid | 3.31 | 0.44 | 0.084 | 0.28 | 11.97 |
| Tan 2017 [48] | TP5 | Myristic Acid | 1.73 | 0.32 | 0.084 | 0.15 | 11.86 |
| Tang 2017 [40] | 22A | synthetic HDL | 2.46 | 14.38 | 0.039 | 0.09 | 25.92 |
| Zorzi 2017 [41] | UK18 | Peptide-FA tag | 7.10 | 0.16 | 0.085 | 0.61 | 11.70 |

respectively) used in the studies. As seen in Fig 3B, the 95% CI of the combined effect size estimate does not cross the line of null effect indicating that the improvement in half life is statistically significant. Additionally, the combined effect estimate is contained on the right side of the line of null effect (favouring conjugation). This indicates that the combined effect is statistically significant and favours conjugation, suggesting that acylation of peptide drugs contributes to a statistically significant increase in mean drug half-life.

**Sensitivity analysis.** To further understand the high heterogeneity in the studies, a sensitivity analysis of the influence of individual studies on the combined effect was conducted (Fig 4). To do so, the combined effect size was compared when individual studies were excluded from the previous random effects model [27]. While there was more than one study that skewed the results (Fig 4), it seems that Zorzi *et al.* [41] had the most impact on the combined effect size as seen by the deviation recorded in Fig 4. This deviation may be attributed to the differences in sample size and peptide used in the Zorzi *et al.* study. More significantly, this deviation could also be due to the specifically synthesized acylated recombinant peptide conjugate that led to a larger effect size (ES = 7.10; Table 3) as compared to the other studies we analysed [39]. Zorzi *et al.* describes this peptide-FA tag as an HSA-binding heptapeptide acylated with palmitic acid. This sensitivity analysis, therefore, suggests that specifically engineered conjugate moieties may contribute to an increase in therapeutic peptide half-life.

**Risk of bias.** A Risk of Bias analysis is a tool commonly used in clinical studies to assess the methodological criteria of a study and their likelihood of introducing systematic errors in the direction or magnitude of the results [28]. We chose to assess inter-study risk of bias (RoB) using the SYRCLE RoB tool (Fig 5). A breakdown of the RoB analysis for each individual study is provided in S3 Table. RoB was analysed across several domains related to the methods and reporting of the experiments in the studies. However, several studies included in our SRMA did not adequately detail methodology of interest in pre-clinical trials such as details on sequence generation, anonymization methods, and random outcome assessment. This made the RoB analysis challenging. Apart from domains like 'Baseline Characteristics' and 'Selective Reporting,' RoB was recorded as 'unsure' for many domains. Of the studies that did provide

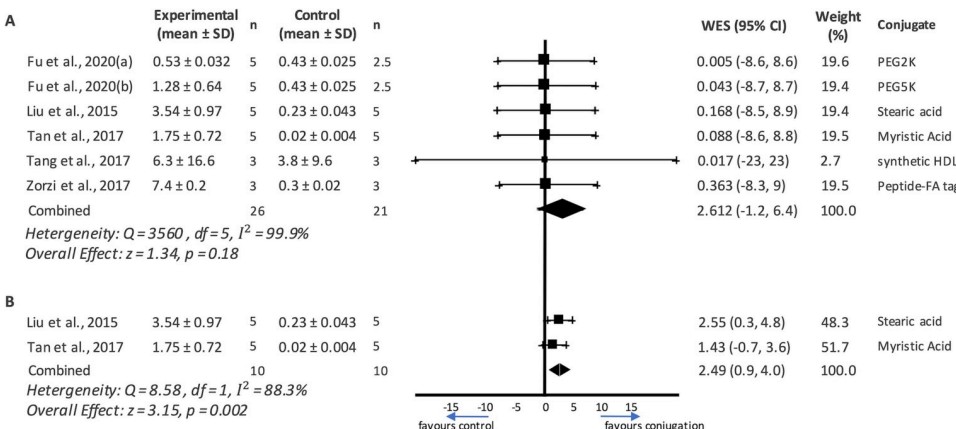

**Fig 3.** Forest plots detailing summary statistics from (A) all the meta-analysed peptides and (B) peptides that use fatty acids as conjugates. The squares indicate the point estimate of the weighted effect size (WES) of individual studies. The size of the square is indicative of the weight assigned to the study. The diamond at the bottom indicates the combined effect size and 95% confidence intervals. A vertical line through the vertices of the diamond reads the point estimate of the combined effect whereas a horizontal line through the vertices indicates the 95% confidence interval. Forest Plot A indicates very high heterogeneity ($I^2 = 99.9\%$) and is not statistically significant (p = 0.18). Forest Plot B indicates high heterogeneity ($I^2 = 88.3\%$) but is statistically significant (p = 0.002). SD = standard deviation, WES = weighted mean difference.

## Sensitivity Analysis

Fig 4. **Sensitivity analysis for the influence of individual studies on the combined effect.** The central thick line indicates the overall effect as seen in Fig 3. The two vertical lines on either side indicates its 95% CI. Each diamond indicates the combined effect observed when the study on the left is omitted. The crosses on either side represent its respective 95% CI.

adequate detail, there were a few that scored high and low RoB in certain domains. The reasoning for a high risk of bias score for studies captured in our meta-analysis are as follows. Fu *et al.* [45] recorded high RoB in the 'Baseline Characteristics' domain as it only mentioned the

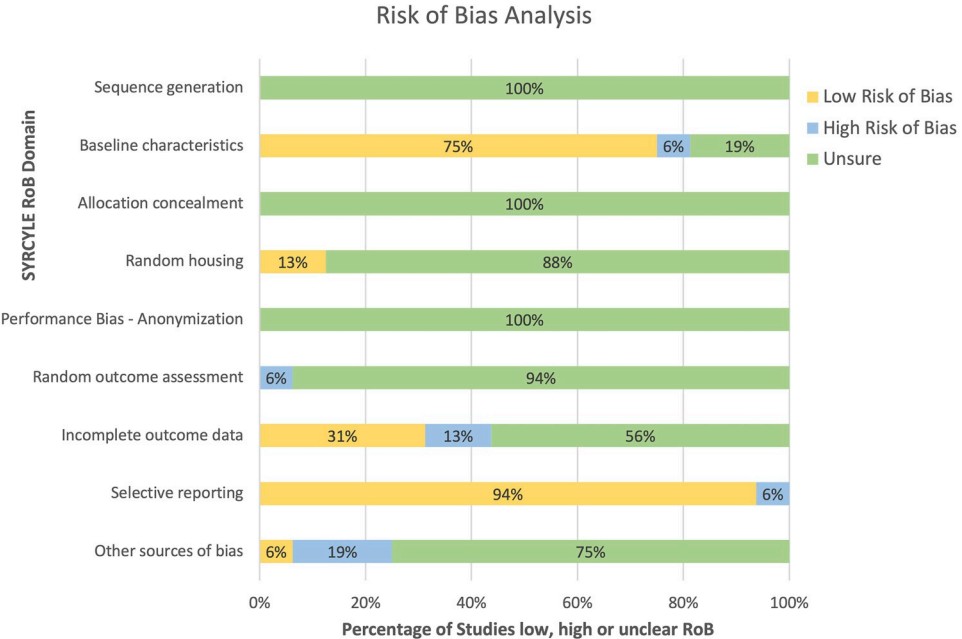

Fig 5. **Risk of bias analysis conducted for the included studies.** SYRCLE's Risk of Bias (RoB) domains were used to assess risk of bias in the studied included in the SRMA. The modifications made to this template are noted in the materials section. The breakdown of RoB analysis for each individual study is provided in S3 Table.

sex and weight of the animals. There was no mention of the animal's level of physical health, although we presumed the animals were healthy based on the rest of the information provided. Similarly, Tang *et al.* [40] recorded high RoB in 'Incomplete Outcome Reporting' because of an unexplained attrition in sample size reported in the results as compared to that reported in the methods. All 5 studies meta-analysed scored a majority of unsure and low RoB in the remaining domains. To sum up, a complete RoB is not possible due to the lack of detail reported.

## Conclusion

The purpose of this systematic review and meta-analysis was to identify the various peptide conjugate moieties currently under investigation, and which of them conferred the greatest plasma half-life to peptide therapeutics. Of the conjugates included in this review, glycosaminoglycans (HPN and CH) were the most effective in terms of increasing plasma half-life as measured by the absolute mean difference between conjugated vs unconjugated peptides (Fig 2). Further, in the five eligible studies that were meta-analysed, an acylated recombinant peptide that was engineered specifically for the therapeutic peptide substantially increased the plasma half-life of the therapeutic. Additionally, in comparing fatty acid groups as conjugates (Fig 3B), a statistically significant combined effect size was observed, indicating that half-life increased when the peptide is acylated. This supports our observation that fatty acids are effective conjugates in increasing plasma half-life. However, it is important to bear in mind that the peptide itself and how it interacts with its surroundings also plays a role in its plasma half-life.

When surveying the studies involved in this SRMA, we noticed several details that were not consistent with the expectations for robust pre-clinical trials detailed by SYRCLE. The absence of control data might be seen as one such noteworthy limitation of the studies included. Having a control treatment in the same experiment is crucial as it allows for unbiased comparisons. Some therapeutic peptides may, however, have half-lives that are so short that measuring the half-life of the unconjugated peptide within the time intervals used for the conjugated form may be challenging. One way to work around this issue may be to include time points with shorter intervals immediately after administration. This could possibly capture the half-life of the rapidly eliminated unconjugated peptide alongside that of the conjugated form. Furthermore, the animal model used in most of the studies involved only male animals. This may create issues in extrapolating results to the entire population of animals and subsequently humans. Newer guidelines suggest using a mix of sexes in the experimental group to identify any important differences in how drugs might affect the different sexes [42]. Knowing these differences provides insight on how the drug might ultimately affect humans. Many the studies included in the review were also assigned an 'unsure' RoB grading in many domains including sequence generation, allocation concealment, random housing, performance bias, and random outcome assessment (Fig 5). This indicates that there was not enough detail reported to make a fair RoB assessment in that particular domain. Including more details when reporting could increase these drugs' likelihood of proceeding into clinical testing. Conversely, a lack of reported details could lead to issues with reproducibility and translatability preventing further development of the therapeutic. This translational gap between basic science discoveries and clinical drug development has come to be known as the 'valley of death' in drug development [49]. Many of these highlighted limitations could nonetheless be avoided by establishing and adhering to comprehensive animal study reporting guidelines as is typical of human trials.

Overall, this systematic review highlighted the large variety of conjugates being studied for use in therapeutic peptides. Additionally, it brought to light a few factors that may contribute to the translational gap in the field of peptide therapeutics. While there is no doubt that

peptides will proceed to hold an important place in pharmaceuticals and healthcare, we are hopeful that future translational research will expedite the process of incorporating these peptide therapeutics into general use against a plethora of diseases.

## Supporting information

**S1 Checklist. PRISMA checklist.**
(DOCX)

**S1 Table. Systematic search strategy used for PubMed, Scopus, and SciFinder.**
(PDF)

**S2 Table. Characteristics of relevant studies captured in the systematic search.**
(PDF)

**S3 Table. Breakdown of the risk of bias analysis for individual studies.**
(PDF)

**S1 Protocol. PROSPERO protocol registration.**
(PDF)

**S1 Data. Excel spreadsheet with all numerical data and statistical formulae.**
(XLSX)

## Acknowledgments

Special thanks to Kristen Romme and Erin Alcock from the Memorial University Libraries for their invaluable contribution in tuning up our search strategy, the backbone of this research.

## Author Contributions

**Conceptualization:** Ashan Wijesinghe, Valerie Booth.

**Data curation:** Ashan Wijesinghe.

**Formal analysis:** Ashan Wijesinghe.

**Funding acquisition:** Valerie Booth.

**Investigation:** Ashan Wijesinghe, Sarika Kumari, Valerie Booth.

**Methodology:** Ashan Wijesinghe, Valerie Booth.

**Supervision:** Valerie Booth.

**Writing – original draft:** Ashan Wijesinghe.

**Writing – review & editing:** Sarika Kumari, Valerie Booth.

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
