## [Decision Letter · Decision Letter 0]

27 Dec 2021

PONE-D-21-23729Conjugates for use in peptide therapeutics: a systematic review and meta-analysisPLOS ONE

Dear Dr. Booth,

Thank you for submitting your manuscript to PLOS ONE. After careful consideration, we feel that it has merit but does not fully meet PLOS ONE’s publication criteria as it currently stands. Therefore, we invite you to submit a revised version of the manuscript that addresses the points raised during the review process. Your manuscript was well written and liked by both reviewers. Some minor corrections are suggested by the reviewers.

We look forward to receiving your revised manuscript.

Kind regards,

Aldrin V. Gomes, Ph.D.

Academic Editor

PLOS ONE

Journal Requirements:

Reviewers' comments:

Reviewer's Responses to Questions

**Comments to the Author**

1. Is the manuscript technically sound, and do the data support the conclusions?

Reviewer #1: Yes

Reviewer #2: Yes

2. Has the statistical analysis been performed appropriately and rigorously? 

Reviewer #1: I Don't Know

Reviewer #2: Yes

3. Have the authors made all data underlying the findings in their manuscript fully available?

Reviewer #1: Yes

Reviewer #2: Yes

4. Is the manuscript presented in an intelligible fashion and written in standard English?

Reviewer #1: Yes

Reviewer #2: Yes

5. Review Comments to the Author

Reviewer #1: This systematic review and meta-analysis examined the available literature (2015-2020) and assessed it objectively to identify which biological and synthetic conjugates produce the largest improvement in therapeutic peptide half-life. Their search has been done in 3 databases, a total of 16 articles were included in the review and 5 in the meta-analysis part. This article appears to be appropriate for publishing, but there are a few problems that need to be corrected.

1. On page 5 the search date is until December 15, 2020, but on page 6 it is September 1, 2020.

2. In Prisma chart, included studies should be written 16, not 19.

Reviewer #2: Based on the requirements that PLOS ONE has for meta-analysis and systematic review publications, your paper does not mention the grant that was listed in the financial disclosure. PLOS ONE requires that any funding of the research be mentioned in the publication, as well as the role of the funders if they have one, as noted in their PRISMA 2009 Checklist for systematic reviews and meta-analysis submission guidelines.

6. PLOS authors have the option to publish the peer review history of their article (what does this mean?). If published, this will include your full peer review and any attached files.

Reviewer #1: No

Reviewer #2: No

---

## [Author Response · Author response to Decision Letter 0]

3 Jan 2022

We would like to thank the editor and the reviewers for the time and thought put into reviewing our manuscript. The work has been revised in response to the comments. 

We have left the reviewer’s comments in black font and have added our point-by-point response in blue (colours visible on attached response file). 

Reviewer #1: This systematic review and meta-analysis examined the available literature (2015-2020) and assessed it objectively to identify which biological and synthetic conjugates produce the largest improvement in therapeutic peptide half-life. Their search has been done in 3 databases, a total of 16 articles were included in the review and 5 in the meta-analysis part. This article appears to be appropriate for publishing, but there are a few problems that need to be corrected.

1. On page 5 the search date is until December 15, 2020, but on page 6 it is September 1, 2020.

2. In Prisma chart, included studies should be written 16, not 19.

Apologies for the confusion with dates. The date on page 5 (11 December 2020), was the date that we last searched the three databases to extract studies relevant to our review. It is a PRISMA requirement that the date that each database was last searched be specified – so this information was included. The date range mentioned on page 6 (1 Sept 2015 to 1 Sept 2020) is what we used to restrict the search to capture only studies published/made available online in last 5 years (rationalized in the paper). The paragraph in page 5 was adjusted as follows: 

“…Where possible, the search was also limited by publication language (English) and date range (1st Sept 2015 to 1st Sept 2020) to reflect the characteristics highlighted in the eligibility criteria (Table 2). …The databases were last searched on 11 Dec 2020. “

The PRISMA flowchart has been adjusted accordingly. 

Reviewer #2: Based on the requirements that PLOS ONE has for meta-analysis and systematic review publications, your paper does not mention the grant that was listed in the financial disclosure. PLOS ONE requires that any funding of the research be mentioned in the publication, as well as the role of the funders if they have one, as noted in their PRISMA 2009 Checklist for systematic reviews and meta-analysis submission guidelines.

The grant specified in the financial disclosure and the role of the funders has been added to the acknowledgements section of the manuscript as follows: 

“This work was supported by a Discovery Grant from the Natural Sciences and Engineering Research Council (NSERC) of Canada to Valerie Booth (RGPIN 05154). The funding source was not involved in study design; in the collection, analysis and interpretation of data; in the writing of the report; or in the decision to submit the article for publication.”

---

## [Decision Letter · Decision Letter 1]

15 Feb 2022

Conjugates for use in peptide therapeutics: a systematic review and meta-analysis

PONE-D-21-23729R1

Dear Dr. Booth,

We’re pleased to inform you that your manuscript has been judged scientifically suitable for publication and will be formally accepted for publication once it meets all outstanding technical requirements.

Kind regards,

Aldrin V. Gomes, Ph.D.

Academic Editor

PLOS ONE

Additional Editor Comments (optional):

Reviewers' comments:

Reviewer's Responses to Questions

**Comments to the Author**

1. If the authors have adequately addressed your comments raised in a previous round of review and you feel that this manuscript is now acceptable for publication, you may indicate that here to bypass the “Comments to the Author” section, enter your conflict of interest statement in the “Confidential to Editor” section, and submit your "Accept" recommendation.

Reviewer #1: All comments have been addressed

Reviewer #2: All comments have been addressed

2. Is the manuscript technically sound, and do the data support the conclusions?

Reviewer #1: Yes

Reviewer #2: Yes

3. Has the statistical analysis been performed appropriately and rigorously? 

Reviewer #1: I Don't Know

Reviewer #2: Yes

4. Have the authors made all data underlying the findings in their manuscript fully available?

Reviewer #1: Yes

Reviewer #2: Yes

5. Is the manuscript presented in an intelligible fashion and written in standard English?

Reviewer #1: Yes

Reviewer #2: Yes

6. Review Comments to the Author

Reviewer #1: The authors have edited my comments. This article appears to be appropriate for publishing.

Sincerely yours,

Reviewer #2: (No Response)

7. PLOS authors have the option to publish the peer review history of their article (what does this mean?). If published, this will include your full peer review and any attached files.

Reviewer #1: No

Reviewer #2: No

---

## [Editor Report · Acceptance letter]

21 Feb 2022

PONE-D-21-23729R1 

Conjugates for use in peptide therapeutics: a systematic review and meta-analysis 

Dear Dr. Booth:

I'm pleased to inform you that your manuscript has been deemed suitable for publication in PLOS ONE. Congratulations! Your manuscript is now with our production department. 

Kind regards, 

on behalf of

Dr. Aldrin V. Gomes 

Academic Editor

PLOS ONE